# Utilizing a multi-stage transition model for analysing child stunting in two urban slum settlements of Nairobi: A longitudinal analysis, 2011-2014

Michael S. Oduro[1,2]◉, Samuel Iddi◉[3,4]◉*, Louis Asiedu[4]◉, Gershim Asiki[3]◉, Damazo T. Kadengye[3,5]◉

1 Pfizer, Inc., Pharm Sci and PGS Statistics, Groton, Connecticut, United States of America, 2 Department of Applied Statistics and Research Methods, University of Northern Colorado, Greeley, Colorado, United States of America, 3 Research Division, African Population and Health Research Center (APHRC), Nairobi, Kenya, 4 Department of Statistics and Actuarial Science, University of Ghana, Legon, Accra, Ghana, 5 Department of Economics and Statistics, Kabale University, Kabale, Uganda

◉ These authors contributed equally to this work.
* siddi@ug.edu.gh; siddi@aphrc.org

⊙ OPEN ACCESS

**Data Availability Statement:** The data used for this research is owned by the African Population and Health Research Center (APHRC) and is available upon request through the Center's microdata portal

## Abstract

### Introduction

Stunting is common among children in many low and middle income countries, particularly in rural and urban slum settings. Few studies have described child stunting transitions and the associated factors in urban slum settlements. We describe transitions between stunting states and associated factors among children living in Nairobi slum settlements.

### Methods

This study used data collected between 2010 and 2014 from the Nairobi Urban and Demographic Surveillance System (NUHDSS) and a vaccination study nested within the surveillance system. A subset of 692 children aged 0 to 3 years, with complete anthropometric data, and household socio-demographic data was used for the analysis. Height-for-age Z-scores (HAZ) was used to define stunting: normal (HAZ $\geq$ 1), marginally stunted (-2 $\leq$ HAZ < -1), moderately stunted (-3 $\leq$ HAZ < -2), and severely stunted (HAZ < -3). Transitions from one stunting level to another and in the reverse direction were computed. The associations between explanatory factors and the transitions between four child stunting states were modeled using a continuous-time multi-state model.

### Results

We observed that 48%, 39%, 41%, and 52% of children remained in the normal, marginally stunted, moderately stunted, and severely stunted states, respectively. About 29% transitioned from normal to marginally stunted state, 15% to the moderately stunted state, and 8% to the severely stunted state. Also, 8%, 12%, and 29% back transitioned from severely stunted, moderately stunted, and marginally stunted states, to the normal state,

which can be reuested and assessed using this link: http://microdataportal.aphrc.org/index.php/catalog.

**Funding:** The author(s) received no specific funding for this work.

**Competing interests:** The authors have declared that no competing interests exist.

respectively. The shared common factors associated with all transitions to a more severe state include: male gender, ethnicity (only for mild and severe transition states), child's age, and household food insecurity. In Korogocho, children whose parents were married and those whose mothers had attained primary or post-primary education were associated with a transition from a mild state into a moderately stunted state. Children who were breastfed exclusively were less likely to transition from moderate to severe stunting state.

## Conclusion

These findings reveal a high burden of stunting and transitions in urban slums. Context-specific interventions targeting the groups of children identified by the socio-demographic factors are needed. Improving food security and exclusive breastfeeding could potentially reduce stunting in the slums.

## Introduction

Stunting, a form of growth impairment observed mostly among children under the age of 5 years, is a pervasive public health problem in many developing countries. It is defined as a child who has low height for his or her age. In 2017, global child stunting prevalence was 22.9% representing 155 million children, with 59 million and 87 million observed in Africa and Asia respectively [1]. Stunting is associated with higher child mortality, poor cognitive functioning, and low educational performance particularly in primary and adolescent school years [2, 3]. Later in adulthood, stunting can lead to poor wages and productivity loss, and an increased risk of chronic diseases. For example, after accounting for other risk factors, a 20% income disparity was found between adults who were stunted in childhood and those who were not stunted [4].

Due to these debilitating consequences of child stunting, addressing stunting is crucial, and has been included as one of the critical components of the Sustainable Development Goals [5]. The WHO aims to decrease stunting globally among children under 5 years of age by 40% by 2025 [5, 6]. This target may be difficult to achieve in sub-Saharan Africa considering that only a 6% reduction (from 38% to 32%) is estimated by 2025 [7]. The past two decades have seen a considerable reduction in the number of stunted children in Latin America and the Caribbean, where in particular, stunting has declined twice as quickly as in Africa from 2000 to 2016 [7]. Child stunting is a consequence of a broad range of factors. Fenske, et. al. [8], Saxton et. al. [9], Shrimpton & Kachondham, [10] and several other researchers have sought to examine underlying determinants and risk factors of child stunting under different paradigms. Some of these determinants are non-modifiable [8, 11], such as sex and age. Male infants, for instance, have a higher risk to be stunted in the first 12 months relative to their female peers, whose vulnerability increases after 24 months [3]. The marked difference in vulnerability is greatly influenced by parental care patterns and cultural norms [12].

Immediate, modifiable risk factors include insufficient maternal and child nutrient intake [13] and poor hygienic practices which are important drivers of stunting. Since stunting mostly happens between 6 months to 2 years, adequate nutrition, involving a mix of complementary feeding and continuous breastfeeding [14] is vital for effective physical growth and cognitive functioning in children during these periods [15]. An additional area of intervention is to accelerate the use of multiple micronutrient supplementation (MMS) for pregnant

women. MMS has been shown to be effective in addressing stunting among children in several contexts worldwide [16–18]. However, access to nutritious foods for complementary feeding might be limited due to financial constraints, geographic location, or availability of diverse food options, affecting the quality of the child's diet. Furthermore, challenges such as the lack of support, maternal employment, and family misconceptions and/or influences about breast-feeding might lead to early cessation of exclusive breastfeeding or reduced frequency of breast-feeding [19, 20]. Also, Multiple micronutrient supplements can be expensive, making them inaccessible for vulnerable populations in low-income settings. This cost factor can limit the effectiveness of interventions aimed at improving nutritional status. Also, ensuring a consis-tent and reliable supply of multiple micronutrient supplements, especially in remote or resource-constrained areas, can be difficult due to logistical issues and infrastructure limita-tions [21, 22].

Lack of healthcare, sanitation and water services [10], household food insecurity, recurrent infections, maternal education, and urban or rural residency [6] have been broadly identified as some of the leading causes of stunting in children. For instance, mothers with lower educa-tional status and income have a relatively higher likelihood of having stunted children, owing to their inability to seek related information on stunting and to afford a balanced, nutritious diet for their infants [23–25]. There is mounting evidence to suggest an association between poor water, sanitation and hygiene (WASH) practices and child stunting. Children living in polluted environments are mostly subject to infections such as diarrhoea, malaria and respira-tory illnesses, which are significant determinants of stunting [9, 15]. Studies conducted in Ghana, Brazil, Peru, and Guinea Bissau have demonstrated that diarrhoeal infections resulting from unhygienic environments, seem to have broader implications on the likelihood of stunt-ing in children by age 2, due to associations with nutrient malabsorption and reduced appetite [7, 26].

In many developing countries, stunting is more prevalent in rural children than in those who live in urban areas. This, in part, can be attributed to the relative ease of access to better health care, health policy programs and WASH services in urban areas [27–29]. However, stunting prevalence is higher among children in low socioeconomic status (SES) urban house-holds than those from households of high socioeconomic status in urban areas. For example, a study conducted in 11 developing countries from South East Asia, South America and Africa identified the risks of urban child stunting in children of low SES as ten times higher than those living in urban areas with higher SES [30]. Thus, targeted interventions, ranging from nutritional programs to education campaigns are needed to ameliorate child stunting not only in rural areas but also in urban areas of low SES. East African countries generally experience higher levels of child stunting compared to other developing countries. While a 14% decrease in stunting overall was observed in developing countries over the past two decades, a 2% increase has been observed in Eastern Africa [31].

In Kenya, for example, between 30 to 40 percent of children under age 5 are stunted, as per national survey estimates [32]. Among this population, disparities are observed between rural and urban child dwellers. Specifically, among the older group of children, living in the urban areas, compared with rural areas, was associated with higher odds of underweight in Kenya [33]. In addition, between 2006 and 2010, about 50% of children under age 5, living in infor-mal urban settlements in Nairobi, Kenya's capital, were considered stunted [34]. Examples of these rapidly expanding urban slum settlements are Korogocho and Viwandani, situated a few miles away from the Nairobi city center. These slums are characterized by poor sanitary condi-tions, abject poverty and limited or no access to health care services. Children born into and living in these settlements are at an increased risk of morbidity [35], or varying levels of stunt-ing (from being marginally stunted to severely stunted) due to exposure to multiple health

hazards [36]. This poses a serious public health concern and necessitates various targeted health interventions. It is thus imperative to investigate underlying factors associated with child stunting in these urban slum settlements to inform public health interventions. Stunting severity may vary markedly across children, and along their life course. Thus, an exploration and/or inference of transitions across different severity stages and associated risk factors would benefit intervention decisions.

Few studies have attempted to investigate child stunting transitions in urban slum settlements and the factors associated with them. Using longitudinal datasets from the Nairobi Urban Health and Demographic Surveillance System (NUHDSS), we attempt to, first of all, explore transitions between stunting states of children living in urban slum settlements. A second objective is to investigate factors that are associated with these movements from one stunting state to another.

## Materials and methods

### Data description

The data for this study were obtained from two main sources, namely, the Nairobi Urban Health and Demographic Surveillance System (NUHDSS) and the INDEPTH Vaccination Project (IVP). The NUHDSS is a longitudinal observational study that gathers data on residents and households within two of Nairobi's informal settlements, Viwandani and Korogocho. Since 2002, the surveillance system has gathered data on health outcomes such as morbidity, cause of death, nutrition, vaccination, births, deaths, and migration. Other data include education, livelihood, and housing, among others. Detailed information about the study design and data collection processes was published elsewhere [37]. The NUHDSS also serves as a platform for nesting other studies. One such study is the IVP study that was designed to monitor and assess the impact of vaccination and interventions on children across six Health and Demographic Surveillance Systems (HDSS) sites within the INDEPTH network funded by the Danish Development Agency [38]. The study was conducted from 2010 to 2014, with data collection starting in March 2011 to June 2014 on children below three years of age. Consequently, outcomes were assessed for some children during follow-up when they reached 40 months, specifically for those who entered the study at 2 to 3 years of age. Among the various data collected on each child during each visit (at most three per child per year) were the child's anthropometric measures (height, weight) and age. Additionally, maternal data on age, education, and marital status were collected.

In this study, data on 692 children with complete anthropometric information, obtained from the IVP study, and whose household socio-economic data were obtained from the NUHDSS were used for the analysis. The main outcome of the study is child stunting measured using height-for-age Z-scores (HAZ) and is derived by comparing child's growth measurements against growth data from the World Health Organization (WHO) international growth reference database [7]. As per WHO global child growth standards, children whose HAZ scores are 2 standard deviations less than the median growth standard are considered stunted [7]. The outcome has four levels, namely, normal (HAZ$\geq 1$), marginally stunted ($-2 \leq$ HAZ$< -1$), moderately stunted ($-3 \leq$ HAZ$< -2$), and severely stunted (HAZ$< -3$). While marginal stunting does not fall into the strict definition of stunting, it cannot be ignored as it represents a barrier to thriving [39].

In addition to the stunting state, we compute the time (in months) it took for a child to transition from one state to another. Explanatory factors considered for this study include the child's gender, child age, breastfeeding status, maternal age, marital status, education, and ethnicity. Household-level factors include household wealth status, food security status, and

household access to water and sanitation. These variables are used to assess the effect of the various factors on transitions between different stunting states of children.

## Ethics statement

The NUHDSS and IVP study were granted ethical clearance by the Ethical Review Board of the Kenya Medical Research Institute (KEMRI). Anonymized dataset was obtained from the African Population and Health Research Center (APHRC) microdata portal. Thus, no risk of harm is posed to the study participants.

## Statistical analysis

In this study, the association between explanatory factors and the transitions between four child stunting states are modeled via a continuous-time multi-state model. The hypothesized 4-state transitions are depicted in Fig 1. More broadly, a key assumption of this model is that individual transitions between states are independent and follow a continuous-time stochastic process. Time is considered homogeneous, such that the assumption of a constant transition rate between states is plausible. A terse mathematical description of the multi-state model is given. Generally, a random, independent, individual movement across a state, $w$ follows a continuous-time stochastic process $\{G(t):t \geq 0\}$, with state space, $s$. If it follows the Markov property, then for $0 \leq s \leq t$, we can obtain a $w \times w$ transition probability matrix, $A(u, v)$, as

$$a_{uv}(s, t) = P(G(t) = u \mid G(s) = v), \quad \text{for } u, v = 1, 2, 3, 4. \tag{1}$$

This reflects the idea that transition probabilities are independent of the past history or process prior to time $t$. Associated transition intensities governing the probabilities can be obtained via the quantity

$$\vartheta_{uv}(t) = \lim_{\Delta t \to 0} \frac{A_{uv}(t + \Delta t, t)}{\Delta t}, \quad u \neq v \tag{2}$$

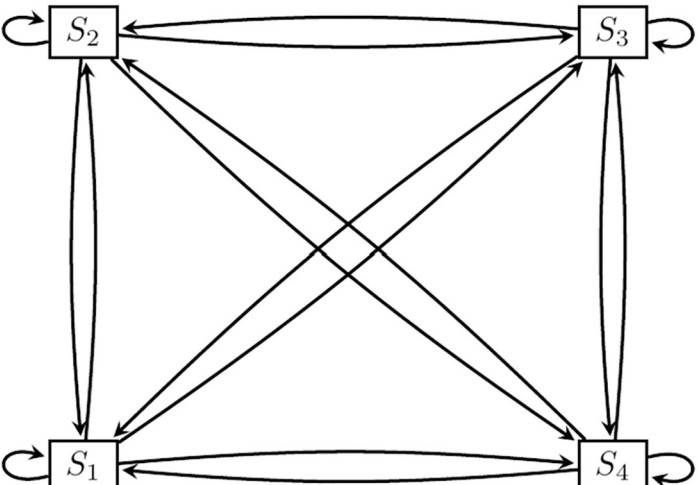

**Fig 1. Hypothesized 4-state transitions.** *States*: $S_1$: Normal, $S_2$: Marginally stunted, $S_3$: Moderately stunted, $S_4$: Severely stunted.

The estimated transition intensities define the instantaneous risk of moving from state $u$ to $v$ and form a transition intensity matrix $R(t)$. Each row in the intensity matrix $R(t)$ needs to sum up to zero and the resulting diagonal entries can be obtained as,

$$\vartheta_{uu} = -\sum_{v \neq u} \vartheta_{uv} \tag{3}$$

The transition or sojourn rate in the continuous-time Markov process has an exponential distribution with rate $-\vartheta_{uu}$ and mean sojourn rate $-1/\vartheta_{uu}$. An individual move from state $u$ to $v$ has a probability of $-\vartheta_{uv}/\vartheta_{uu}$. Varying assumptions about the dependence of the transition intensities on time can be made and since a time homogeneous model is considered in our study, we consider a constraint, $\vartheta_{uv}(t) = \vartheta_{uv}$ on time. Incorporating covariate information in the continuous-time, time homogeneous Markov model is plausible. Both time-independent and time-varying variables can be included, via a specified proportional hazards regression model, represented by:

$$\vartheta_{uv}(\boldsymbol{X}(\boldsymbol{t})) = \vartheta_{uv,0}\exp(\alpha_{uv}^T \boldsymbol{X}(\boldsymbol{t})) \tag{4}$$

Here, $\vartheta_{uv,0} = \exp(\alpha_{uv,0})$ represents the baseline transition intensity from state $u$ to $v$. $\alpha_{uv}^T$ is the coefficient vector of the covariates on the transition intensities. Also, $\exp(\alpha_{uv})$ reflects the rate or hazard ratio of intensity for the selected covariates, which explains the instantaneous risk of transition. Specifically, we include demographic variables on gender, ethnicity, child age, food security, mother's marital status, and education. Other factors considered in the model are proper sanitation access, breastfeeding status, safe water access, and the wealth status of households. Parameter estimation of the multi-state model considered hinges on Maximum Likelihood. Extensive information on the estimation process can be found in [40]. The analysis is implemented in R software via the package msm [41].

## Results

### Descriptive analysis

Descriptive statistics of the outcome and each predictor variable at entry to the study across the recruitment years, 2010-2013 are shown in Table 1. From this table, we observed that at least 59% of children were from Korogocho, across the different years. The proportion of children who were male increased from 44% to 78% from 2011 to 2013. In terms of ethnicity, the majority of the children were of the Kikuyu ethnic group representing at least 33% in 2011 and 2012. The proportion of 'Other' ethnic groups that are not part of the four major ethnic groups (Kikuyu, Luhya, Luo, and Kamba) decreased from 18% to 11% across the years. The majority of the children were between the ages of 12-23 months, increasing from 38% in 2011 to 72% in 2013. Furthermore, most of the households in the sample were severely food insecure, with 59% in 2011 and 78% in 2013. We further observed that at least 78% of mothers were ever married, and at least 67% had primary education across the years. Fewer mothers (about 1%) practiced exclusive breastfeeding. Few households (about 1%) had proper sanitation, and 4% had access to safe water. The percentage of households in the lowest wealth tertile was 37% compared to 32% for middle and 31% for highest wealth tertile.

In terms of the main outcome, stunting status, the percentage of normal children in the sample ranged from 21% in 2011 to 28% in 2013. Approximately 29% across the years were marginally stunted, and severely stunted children ranged from 21% in 2011 to 33% in 2013.

Table 2 shows the percentage of observed transitions from one stunting state to another between 2011 to 2014. About 48%, 39%, 41%, and 52% of the observed times, children remained in the normal, marginally stunted, moderately stunted and severely stunted states,

**Table 1. Descriptive statistics at entry into the study (N = 676). N is the number of children.** Test used: Pearson test.

| | N | 2011 | 2012 | 2013 | Combined | P-value |
|---|---|---|---|---|---|---|
| | | N = 432 | N = 226 | N = 18 | N = 676 | |
| Stunting | 676 | | | | | 0.671 |
| Marginally stunted | | 30% | 28% | 22% | 29% | |
| Moderately stunted | | 28% | 25% | 17% | 27% | |
| Normal | | 21% | 24% | 28% | 22% | |
| Severely stunted | | 21% | 23% | 33% | 22% | |
| Slum | 676 | | | | | 0.379 |
| Korogocho | | 59% | 64% | 67% | 61% | |
| Viwandani | | 41% | 36% | 33% | 39% | |
| Sex | 676 | | | | | <0.001 |
| Female | | 56% | 43% | 22% | 51% | |
| Male | | 44% | 57% | 78% | 49% | |
| Ethnicity | 676 | | | | | 0.612 |
| Kikuyu | | 39% | 33% | 28% | 37% | |
| Luhya | | 13% | 16% | 11% | 14% | |
| Luo | | 13% | 14% | 17% | 14% | |
| Kamba | | 17% | 18% | 33% | 18% | |
| Other | | 18% | 18% | 11% | 18% | |
| Child's age | 676 | | | | | <0.001 |
| 0-5 | | 8% | 16% | 11% | 11% | |
| 6-11 | | 16% | 22% | 17% | 18% | |
| 12-23 | | 38% | 44% | 72% | 41% | |
| 24-36 | | 39% | 18% | 0% | 31% | |
| Household food security | 676 | | | | | 0.268 |
| Secure | | 22% | 19% | 11% | 20% | |
| Moderate | | 19% | 15% | 11% | 17% | |
| Severe | | 59% | 66% | 78% | 62% | |
| Mother's marital status | 676 | | | | | 0.739 |
| Never Married | | 23% | 21% | 17% | 22% | |
| Ever Married | | 77% | 79% | 83% | 78% | |
| Mother's education | 676 | | | | | 0.93 |
| Less than Primary | | 6% | 8% | 6% | 7% | |
| Primary | | 72% | 70% | 67% | 71% | |
| Post Primary | | 22% | 23% | 28% | 23% | |
| Access to proper sanitation | 676 | | | | | 0.475 |
| No | | 99% | 100% | 100% | 99% | |
| Yes | | 1% | 0% | 0% | 1% | |
| Breastfeeding | 676 | | | | | 0.762 |
| 0 | | 99% | 99% | 100% | 99% | |
| 1 | | 1% | 1% | 0% | 1% | |
| Access to safe water | 676 | | | | | 0.166 |
| No | | 95% | 98% | 94% | 96% | |
| Yes | | 5% | 2% | 6% | 4% | |
| Household wealth status | 676 | | | | | 0.027 |
| lowest | | 34% | 41% | 56% | 37% | |
| middle | | 31% | 35% | 28% | 32% | |
| highest | | 35% | 24% | 17% | 31% | |

**Table 2. Observed transitions between height-for-age classification.**

| From | To | | | |
|---|---|---|---|---|
| | Height-for-age classification | | | |
| | Normal | Marginally | Moderately | Severely |
| | Z > −1 | −2 < Z < −1 | −3 < Z < −2 | Z < −3 |
| Normal | 256 (48%) | 155(29%) | **77(15%)** | **40 (8%)** |
| Marginally stunted | 175(29%) | 232(39%) | 150(25%) | **44(7%)** |
| Moderately stunted | **71(12%)** | 142(24%) | 249(41%) | 140(23%) |
| Severely stunted | **37(8%)** | **60(13%)** | 128(27%) | 241(52%) |

respectively. About 29%, 15% and 8% of transitions from normal to the marginally stunted state, to the moderately stunted state, and to the severely stunted state, respectively, occurred during the study period. Also, 8%, 12%, and 29% of back transitions from severely stunted, moderately stunted, and marginally stunted states, to the normal state, respectively, occurred during the course of study. Generally, it was observed that the proportion of forward transitions from the normally stunted state to other states decreased. This observation was similarly made for back transitions from other states into the normally stunted state.

Transitions that had percentages less than 10% were not considered in the multi-state modeling due to the small sample size for these transitions which hampers the estimation of model parameters associated with these transitions. Thus, some of the hypothesized transitions were not estimated and the schematic diagram for the 4-state transition shown in Fig 1 reduces to Fig 2. In the next section, we assessed the factors that were associated with transitions from one state of stunting to a more severe state and the reversal of each of these transitions.

## Statistical analysis

**Baseline transitions.** Before fitting the model, the Cramer's *V* statistics were computed to assess any potential correlation between the categorical independent variables. All associations between the independent factors were weak (Cramer's *V* values <0.2) [42], eliminating any

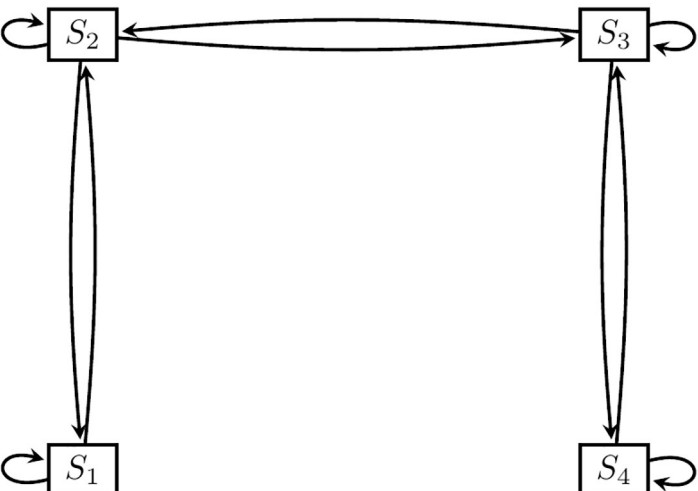

**Fig 2. Fitted transition model.** *States*: $S_1$: Normal, $S_2$: Marginally stunted, $S_3$: Moderately stunted, $S_4$: Severely stunted.

fear of multicolinearity. For example, the Cramer's *V* value for the relationship between food security and wealth quartile was 0.149, between access to sanitation and wealth status was 0.054, and between access to sanitation and food security was 0.088. The baseline transition intensity estimates obtained from the continuous-time multi-state model with and without adjustments for covariates are presented in Table 3, along with their associated 95% confidence intervals. We focus on the results of the model with covariates since it had a better fit, reflected by a lower AIC value of 5281. Inferring from this model, it is observed that children in a marginally stunted state are 42% less likely to move into a moderately stunted state than to a normal state. Furthermore, moderately stunted children have a 49% higher likelihood to transition into a severely stunted state than to a marginally stunted state.

We also estimated the mean sojourn times from the model. On average, children spend about 2.4(95%CI[1.1, 5.4]) months in normal state before moving to other states. Furthermore, the average times spent in the marginally stunted, moderately stunted, and severely stunted state before transitioning to other states are estimated as 1.7(95% CI [1, 3]), 2.1(95%CI[1.4, 3.2]), and 2.27 (95% CI [1.1, 4.9]) months, respectively. In Fig 3, we display the forty-month transition probabilities. We observed relatively higher transition probabilities from severe to moderate, normal to marginal, and marginal to normal stunted states, peaking before 5 months and slowing gradually thereafter.

**Factors associated with each transition state.** Table 4 provides comprehensive information regarding the model estimates and the corresponding statistical significance related to the factors linked with transition states. The subsequent sections will present the inference drawn from these transitions.

**Normal to marginally stunted and backward transitions.** *Child demographic characteristics*. Adjusting for other risk factors, it is observed that male children in the normal state are 4.3 times more likely to transition to the marginally stunted state when compared to female children. For the back transition, male children are observed to be 2.8 times more likely to move back from the marginally stunted state into the normal state when compared to females. Regarding the effect of ethnicity on child stunting, it is observed that the risk of children moving from the normal state to the marginally stunted state in Kamba and other ethnic households is, respectively, 70% and 78% less likely when compared to children from Kikuyu

**Table 3. Baseline transition intensity estimates and corresponding 95% confidence interval from the multi-state model with and without covariates.** State 1: normal, State 2: Marginally stunted, State 3: Moderately stunted, State 4: Severely stunted.

| Transition parameter | Model without covariate | Model with covariates |
|---|---|---|
| State 1—State 1 | -0.2994 (-0.3946,-0.2272) | -0.4214 (-0.9506,-0.1868) |
| State 1—State 2 | 0.2994 (0.2272, 0.3946) | 0.4214 (0.1868, 0.9506) |
| State 2—State 1 | 0.2838 (0.2143, 0.3759) | 0.4114 (0.1831, 0.9243) |
| State 2—State 2 | -0.4874 (-0.5789,-0.4103) | -0.5828 (-1.0219,-0.3324) |
| State 2—State 3 | 0.2036 (0.1692, 0.2450) | 0.1714 (0.1342, 0.2189) |
| State 3—State 2 | 0.2049 (0.1713, 0.2451) | 0.1909 (0.1473, 0.2473) |
| State 3—State 3 | -0.3518 (-0.4029,-0.3071) | -0.4754 (-0.7312,-0.3090) |
| State 3—State 4 | 0.1469 (0.1177, 0.1834) | 0.2845 (0.1365, 0.5929) |
| State 4—State 3 | 0.1960 (0.1587, 0.2420) | 0.4390 (0.2057, 0.9368) |
| State 4—State 4 | -0.1960 (-0.2420,-0.1587) | -0.4390 (-0.9368,-0.2057) |
| -2loglike | 5468.886 | 5041.042 |
| # of Parameters | 6 | 120 |
| AIC | 5480.886 | 5281.042 |

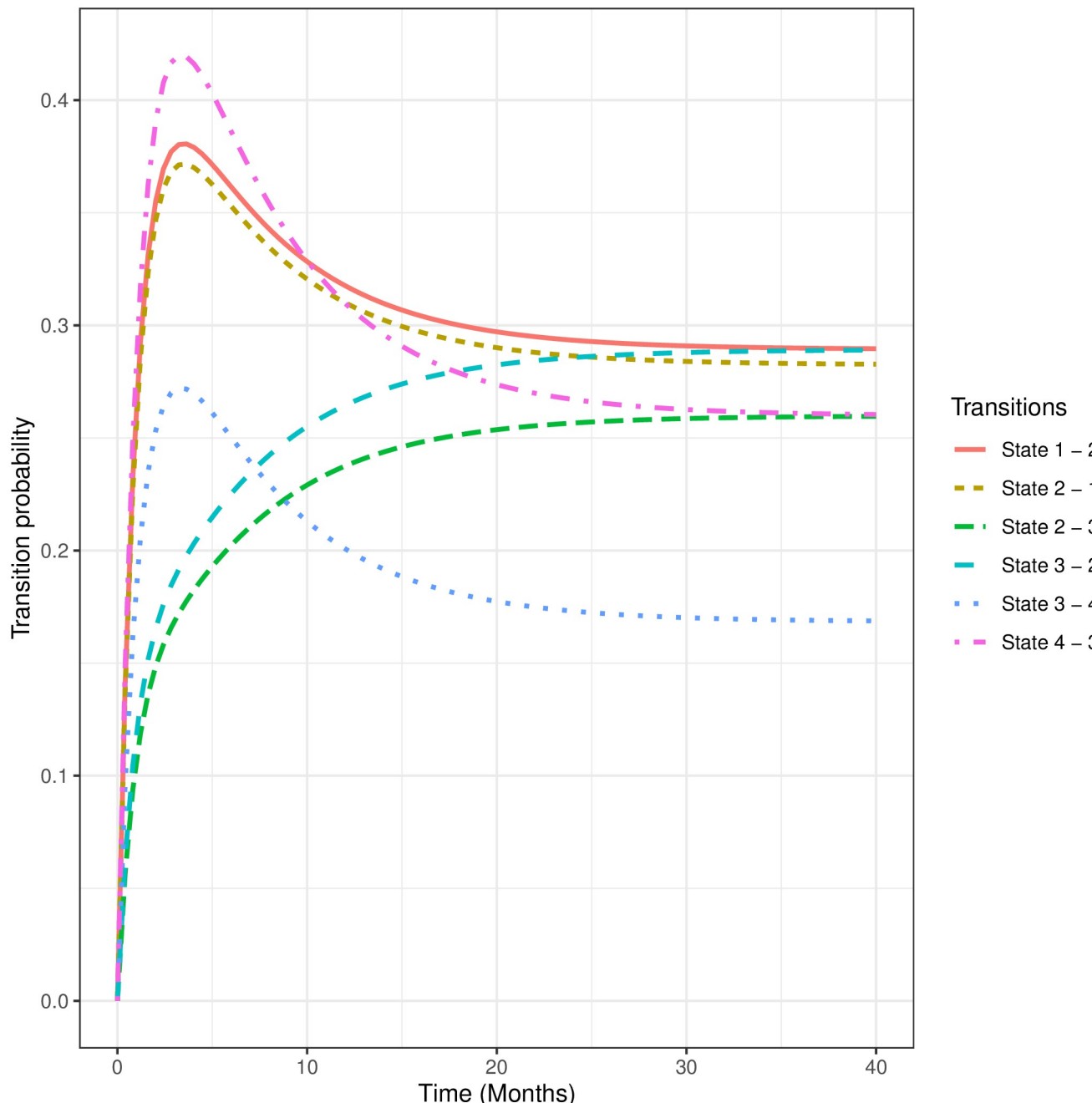

**Fig 3. Forty-months transition probabilities.** State 1: Normal, State 2: Marginally Stunted, State 3: Moderately Stunted, State 4: Severely Stunted.

households. Also, children from Luhya, Luo, Kamba, and other ethnic households are, respectively, 60%, 67%, 71% and 83% less likely to back transition from a marginally stunted to a normal stunted state relative to children from the Kikuyu ethnicity. The results also indicate that, from a normal state, children in the age bracket of 6-11 months and 24-36 months are 67% and 73% less likely, respectively, to transition into a marginally stunted state when compared to infants in the 0-5 month range. When in a marginally stunted state, children within the age

**Table 4. Hazard ratios and corresponding 95% confidence interval from the multi-state model with covariates.** State 1: normal, State 2: Marginally stunted, State 3: Moderately stunted, State 4: Severely stunted.

| Covariates | State 1-2 | State 2-1 | State 2-3 | State 3-2 | State 3-4 | State 4-3 |
|---|---|---|---|---|---|---|
| **Slum site** (ref = Korogocho) | | | | | | |
| Viwandani | 0.6636 (0.3679,1.1970) | 0.6517 (0.3644,1.1654) | **0.3906 (0.2459,0.6205)** | **0.5721 (0.3765,0.8694)** | 1.0727 (0.5546,2.0750) | 1.1876 (0.6221,2.2672) |
| **Gender of HHH** (ref = Female) | | | | | | |
| Male | **4.3388 (2.1560, 8.731)** | **2.8294 (1.4433, 5.547)** | 1.2171 (0.8265, 1.792) | 0.9123 (0.6377, 1.305) | **10.1004 (3.8515,26.488)** | **5.3404 (2.0722,13.763)** |
| **Ethnicity of HHH** (ref = Kikuyu) | | | | | | |
| Luhya | 0.4986 (0.2049,1.2130) | **0.4010 (0.1704,0.9438)** | 1.5979 (0.8069,3.1643) | 1.3803 (0.7145,2.6663) | **0.2942 (0.1050,0.8246)** | **0.2079 (0.0735,0.5874)** |
| Luo | 0.4280 (0.1355,1.3523) | **0.3270 (0.1085,0.9860)** | 0.7886 (0.4417,1.4081) | 0.6555 (0.3820,1.1248) | 0.4186 (0.1439,1.2174) | **0.3113 (0.1085,0.8931)** |
| Kamba | **0.2978 (0.1177,0.7536)** | **0.2879 (0.1214,0.6826)** | 1.2651 (0.7039,2.2737) | 0.9605 (0.5433,1.6980) | 0.3845 (0.1440,1.0268) | **0.2504 (0.0988,0.6345)** |
| Others | **0.2204 (0.0919,0.5288)** | **0.1694 (0.0707,0.4059)** | 0.9023 (0.5211,1.5623) | 0.8475 (0.5017,1.4315) | 0.6895 (0.1825,2.6050) | 0.8723 (0.2179,3.4923) |
| **Child age** (ref = 0-5) | | | | | | |
| 6-11 | **0.3344 (0.1184,0.9440)** | 0.3678 (0.1117,1.211) | 0.4480 (0.1502,1.337) | 0.4016 (0.0986,1.637) | 0.4164 (0.1007,1.723) | 0.5481 (0.1124,2.673) |
| 12-23 | 0.2113 (0.07255,0.6152) | 0.3668 (0.11598,1.1600) | 0.3303 (0.1082,1.0080) | 0.5232 (0.1285,2.1300) | **0.2458 (0.06370,0.9484)** | 0.5050 (0.1141,2.2353) |
| 24-36 | **0.2674 (0.0961,0.7444)** | **0.2623 (0.0854,0.8060)** | **0.1584 (0.051424,0.4877)** | **0.2138 (0.0519,0.8814)** | 0.4268 (0.1134,1.6068) | 0.7870 (0.1824,3.3964) |
| **Hunger scale** (ref = Food secure) | | | | | | |
| Moderately food insecure | 0.8622 (0.3189,2.331) | 0.8376 (0.3262,2.151) | **2.4362 (1.3037,4.552)** | **2.8500 (1.5761,5.154)** | 0.4177 (0.1172,1.489) | 0.9433 (0.2503,3.555) |
| Severely food insecure | 0.3934 (0.1445,1.0708) | **0.3487 (0.1351,0.8997)** | 1.2696 (0.7795,2.0679) | **1.6450 (1.0643,2.5427)** | 0.4033 (0.1313,1.2387) | 0.4642 (0.1423,1.5142) |
| **Marital status** (ref = Never married) | | | | | | |
| Ever married | 0.2273 (0.0167,3.090) | 0.2262 (0.0180,2.845) | **1.7527 (1.1033,2.784)** | 1.2136 (0.7951,1.852) | 0.1731 (0.02829,1.060) | 0.2629 (0.04096,1.687) |
| **Mother's education** (ref = Less than primary) | | | | | | |
| Primary | 1.4427 (0.4640, 4.485) | 0.6502 (0.2533, 1.669) | **2.8513 (1.3584, 5.985)** | **3.0208 (1.4541, 6.275)** | 2.1557 (0.3893,11.938) | 1.9293 (0.3336,11.159) |
| Post primary | 0.9081 (0.26486,3.113) | 0.5582 (0.19497,1.598) | **3.6286 (1.49220,8.824)** | **2.7105 (1.15566,6.357)** | 0.5689 (0.10085,3.209) | 0.3856 (0.06484,2.293) **Access to sanitation** (ref = No) |
| Yes | 4.3811 (0.11683,164.290) | 6.3717 (0.17278,234.974) | 0.4380 (0.05382, 3.565) | 0.8098 (0.09223, 7.109) | 1.4090 (0.07192, 27.605) | 1.0495 (0.04832, 22.795) **Exclusive breastfeeding** (ref = No) |
| Yes | 9.94390 (0.302598, 326.7739) | **61.95368 (1.760710,2179.9489)** | 3.11600 (0.603764, 16.0816) | 0.59146 (0.150893, 2.3184) | **0.06512 (0.004557, 0.9306)** | 0.31491 (0.026592, 3.7293) **Access to safe water** (ref = No) |
| Yes | 0.4063 (0.05711,2.890) | 0.5939 (0.08704,4.052) | 1.7575 (0.42614,7.248) | 1.2255 (0.34979,4.293) | 0.5622 (0.1162,2.720) | 0.6907 (0.1468,3.249) **Household wealth status** (ref = Lowest) |
| Middle | 0.7858 (0.3950,1.563) | 0.9283 (0.4784,1.801) | 1.1226 (0.7171,1.757) | 1.1224 (0.7398,1.703) | 0.6722 (0.3341,1.352) | 0.8734 (0.4378,1.743) |
| Highest | 0.5550 (0.2402,1.283) | 0.5917 (0.2659,1.317) | 1.1380 (0.7105,1.823) | 1.2778 (0.8183,1.996) | 0.6531 (0.2962,1.440) | 0.7676 (0.3493,1.687) |

range 24-36 months are 74% less likely to transition back into the normal state from a marginally stunted state in comparison to children within the 0-5 month range.

*Social determinants of health*. Furthermore, it is observed that children living in severely food insecure households have about a 65% lesser likelihood to back transition from a marginally stunted state to a normal state relative to children in food secured households. Also, children who are exclusively breastfed are 61.95 times more likely to back transition from a marginally stunted state to a normal state compared to children who are not exclusively breastfed. For both forward and backward transitions between normal and marginally child stunting states, we did not find statistically significant effects for slum area, marital status, mother's education, access to sanitation and safe water, and household wealth status.

**Marginally stunted to moderately stunted and backward transitions.** *Child demographic characteristics*. The results in Table 4 further indicate that children in Viwandani households are 61% less likely to transition from a marginally stunted to a moderately stunted state. However, when in a moderately stunted state, children are 43% less likely to back transition into a marginally stunted state compared to those in Korogocho households. It is also observed that the risk of moving into a moderately stunted state from a marginally stunted state is 84% lower for children aged between 24 and 36 months compared to those aged between 0 and 5 months. The likelihood, however, for children aged 24-36 months to back transition from a moderately stunted state to a marginally stunted state is 79% lower than those aged 0-5 months.

*Social determinants of health*. When in a marginally stunted state, children living in moderately food insecure and severely food insecure households are 2.4 times more likely to move into a moderately stunted state, in comparison to children in food secure households. The risk of back transitioning from the moderately stunted to marginally stunted state for children living in moderately food insecure and severely food insecure households are, respectively, 2.9 and 1.7 times the risk for children in food secure households. No statistically significant effect was observed for both forward and backward transitions between marginally stunted and moderately stunted states for access to sanitation services, exclusive breastfeeding, access to safe water, and household wealth status.

*Maternal/paternal characteristics*. We also find that children living in households with parents who have ever married and in a marginally stunted state are 1.75 times more likely to transition into a moderately stunted state, relative to children whose parents have never been married. In addition, when in a moderately stunted state, children whose mothers have attained primary or post primary education are, respectively, 3.02 times and 2.71 more likely to transition into a marginally stunted state compared to children with a less than primary educated mothers.

**Moderately stunted to severely stunted and backward transitions.** *Child demographic characteristics*. The results further show that when in the moderately stunted state, male children are about 10.1 times more probable to transition to a severely stunted state than their female counterparts, and 5.3 times more likely to back transition. Furthermore, children living in Luhya households, when in a moderately stunted state, are at a 71% lower risk to transition into a severely stunted state when compared to those living in Kikuyu households. On the other hand, the likelihood of back transitioning from a severely stunted to a moderately stunted state for children in Luhya, Luo, and Kamba households are respectively 79%, 60%, and 75% lower relative to Kikuyu children. In a moderately stunted state, the risk for children aged 12-23 months to transition into a severely stunted state is 75% lower when compared to children aged 0-5 months.

*Social determinants of health*. In a moderately stunted state, children are 94% less likely to sojourn into a severely stunted state if they are breastfed exclusively, as compared with those

who are not. For both forward and backward stunting transitions between moderate to severe states, we observe no significant effects for slum area, food security status, marital status, access to good sanitation, household wealth status, and access to safe water.

## Discussion

Our study sought to investigate transition dynamics between child stunting states and associated factors in two urban slum settlements of Nairobi, Kenya using a multi-state transition modeling approach. The result of the study indicated that children from Viwandani, when in a marginally stunted state, were less likely to move into a moderately stunted state in comparison to Korogocho children. For the effect of ethnicity on child stunting, we found that moderately stunted children living in Luo, Kamba and Luhya households were at a relatively lower risk of transitioning into normal state when compared to those living in Kikuyu households. There was also a lower risk for them to transition from marginally stunted to normal state. There are conceivable mechanisms that may explain the underlying reasons for the findings presented in this study, and underscore the urgent need for targeted government policies and interventions to address the disparities observed in stunting transitions between the two slum settlements and more broadly, that observed between ethnicities. Viwandani and Korogocho likely exhibit variations in socioeconomic conditions, access to essential resources like clean water, nutritious food, and healthcare services. These disparities might contribute to the differing likelihoods of transitions between stunting states.

Limited access to resources in one settlement could lead to more prolonged stunting states due to inadequate nutrition and healthcare. Furthermore, environmental factors, such as living conditions, sanitation, and exposure to contaminants, can play a significant role in child health. Variations in environmental conditions between the two settlements might influence the progression of stunting states. For instance, poorer sanitation conditions could lead to higher rates of infections, exacerbating stunting.

Disparities in healthcare access and health awareness can impact the likelihood of transitioning out of stunting states. If one settlement or ethnic group has better access to healthcare services and nutrition education, children might have a higher chance of transitioning to a healthier state. Also, differences in government policies and interventions targeting these slum settlements and ethnic groups can also play a role. Variances in the effectiveness and reach of these policies might contribute to the observed disparities in stunting transitions.

Also, in this study, gender effects on stunting transitions were observed. The results showed that male children in a normal state were more likely to transition to a marginally stunted state when compared to female children. There was a similar observation for the male child to transition from a marginally stunted state into a moderately stunted state and from a moderately stunted state to a severely stunted state compared to their female counterparts. This observation is broadly consistent with several research studies [43–46]. For example, results from Wamani, et. al. [43] involving a meta-analysis of 16 demographic and health surveys pointed out that boys are significantly more stunted than girls in sub-Saharan Africa, and it is even more so for those male children in the lowest wealth tertiles. This is further confirmed by Bork & Diallo [45] whose research demonstrated that throughout infancy, boys exhibited lower height-for-age z-scores (HAZs) in comparison to girls, and this gender-related variation became more pronounced up to the age of 39 months. Further, there was a disparity in complementary food intake based on gender, with boys demonstrating a higher likelihood of consuming complementary food.

With regards to the effect of a child's age on different stunting and its transitions, we observed that in a normal state, children within the age range of 6-11 months and 24-36

months were less likely to transition into a marginally stunted state compared to infants in the 0-5 month range. In general, it was observed that children aged 0-5 months were more likely to rapidly transition among the four stunting states compared to the other age groups, regardless of the state they transitioned from. Some research studies [47, 48] corroborate this finding of age being significantly associated with child stunting.

The study of Rakotomanana et. al. [49] which investigates the prevalence of stunting among children in Madagascar under age 5, observes that the risk of stunting generally increases with age, and that stunting determinants are markedly different for children within the age range of 0–23 months and 24–59 months. Furthermore, this finding is in line with Bloss et. al. [50] who inferred via cross-sectional studies that in their second year of life, children in Western Kenya have a higher likelihood to be underweight and stunted when compared with the first year (0–12 months)(OR = 2.34;95% CI, 1.01–5.95). Given the observation that children aged 0-5 months are more likely to rapidly transition among the four stunting states compared to the other age groups, early screening during this period is essential, as it allows for the implementation of targeted and responsive interventions. Whether through nutritional supplementation, breastfeeding support, or health education for caregivers, these interventions can effectively support a child's growth trajectory, minimize long-term consequences, and contribute to overall health and well-being.

Observing food security effect on stunting transitions, we observed that children living in households that experienced moderate to severe food insecurity were at an increased risk to transition into a moderately stunted state from a marginally stunted state, relative to those living in food secure households. This finding aligns with the results of a study by Hackett et. al., [51], which revealed that in Bogota, Colombia, children living in mild, moderate, and severe food insecure households had a significantly higher risk for stunting compared to their counterparts in food secure households. In fact, the odds for young Pakistani children from food insecured households (regardless of the tertile) were 3 times more than those of children from food secured households [52]. This underscores the need for food assistance programs or government policy interventions to bridge the gap between food secure and insecure households. The effect of parent marital status on child stunting transitions was notable. Although not statistically significant, we observed that children living in households with parents who have ever married were less likely to transition to marginally and severely stunted states from a normal state in comparison to those children whose parents have never been married. This finding is similar to results in the study of Blankenship et. al., [53] and Reurings et. al. [54] whose studies found marital status as being significantly associated with child stunting.

Regarding the effect of mother's educational status on child stunting transitions, the results indicated that children in a moderately stunted state were more likely to back-transition to marginally stunted state if they have mothers with primary and post-primary education. This finding is consistent with a study by Abuya et. al. [32], who observed that children born to primary educated mothers were at a significantly lower odds of being stunted relative to mothers with no primary education. Even higher levels of maternal education have been found to significantly reduce the odds of child stunting in Malawi, Tanzania, and Zimbabwe [55]. These results highlight the need for policies to sufficiently educate mothers or the girl child, more broadly, in slums such as Korogocho and Viwandani. Equipping women with more than a primary school education has a significant potential to mitigate the child stunting menace.

Inference for the effect of exclusive breastfeeding on stunting transitions revealed a significantly lowered stunting risk for children in a moderately stunted state to transition into normal state if they were exclusively breastfed, compared to those who were not. There was also a lower risk to transition from a moderately stunted state to a severely stunted state for children who were exclusively breastfed. This is very much in line with study findings of Lestari et. al.

[56] and Beal et. al. [57] who identify exclusive breastfeeding as a protecting factor against stunting among children in Indonesia. As a matter of fact, in their first 6 months of life, it is observed that children who were not exclusively breastfed had a significantly higher chance of being stunted [57]. It is thus imperative that in slum settlements such as Viwandani and Korogocho, extensive public education is provided to mothers to sensitize them on the need and benefits of exclusive breastfeeding, especially in the early months of childbirth.

Albeit not statistically significant, the direction of the effects of household wealth status on transitions was quite insightful. In particular, we observed generally that children living in the highest wealth households had a reduced risk of transitioning into a severely stunted state from a moderately stunted state relative to those from the lowest wealth households. This finding is very much in agreement with Abuya et. al. [32], Agho et. al. [58], and Hong et. al. [59] who studied stunting risk factors in Kenya, Indonesia, and Bangladesh respectively. In Kenya, it was observed that children living in richer households, characterized by a rich wealth index, were 39% less likely to have stunted growth compared to those in poor households [32]. The results are even more telling in Bangladesh, where children in the lowest household wealth tertile were three times more likely to be stunted than children from the wealthiest 20% of households [59].

Taking into account the study's strengths and potential limitations is crucial. Firstly, as far as we know, this study is pioneering in its utilization of a comprehensive multi-state transition modeling approach, enabling the exploration of intricate transitions between various child stunting states and their associated factors. Moreover, this research was conducted within two urban slum settlements in Nairobi, shedding light on a marginalized and often neglected population. Additionally, the inclusion of participants' ethnicity recognizes the sway of cultural elements on child stunting transitions. Nevertheless, it's important to emphasize that the accuracy of the study's outcomes hinges on the quality of the measured data. The existence of measurement errors could potentially impact the conclusions drawn. Moreover, the findings might not be universally applicable to other settings due to the distinct characteristics of the studied slum settlements and ethnic groups. Lastly, it is important to acknowledge the potential presence of unmeasured confounding variables. Socioeconomic indicators and care-giving practices, unaccounted for in this study, could potentially influence the observed associations.

## Conclusion

In conclusion, findings from this study clearly highlight child stunting variations observed between the two urban slums, Korogocho and Viwandani in Nairobi, as evidenced by transitions observed and associated risk factors affecting these transitions. Clearly, stunting transitions are not unidirectional, and certain risk factors tend to increase the likelihood of back transitions into severe stunting states. These factors underscore the need and call for broad, targeted government interventions and policies in these slum settlements. More needs to be done to curb stunting in Kenya's urban slum communities. We recommend change strategies such as improvements in the nutritional status of children, maternal education programs on exclusive breastfeeding, and timely complementary feeding practices in these slum communities.

## Acknowledgments

We sincerely acknowledge those who contributed to the establishment of the NUHDSS, especially Alex Ezeh and Eliya Zulu.

## Author Contributions

**Conceptualization:** Michael S. Oduro, Samuel Iddi, Gershim Asiki.

**Data curation:** Samuel Iddi.

**Formal analysis:** Samuel Iddi.

**Methodology:** Michael S. Oduro, Samuel Iddi, Louis Asiedu, Damazo T. Kadengye.

**Project administration:** Michael S. Oduro.

**Supervision:** Samuel Iddi, Damazo T. Kadengye.

**Validation:** Louis Asiedu, Gershim Asiki, Damazo T. Kadengye.

**Writing – original draft:** Michael S. Oduro, Samuel Iddi, Louis Asiedu, Gershim Asiki.

**Writing – review & editing:** Michael S. Oduro, Samuel Iddi, Louis Asiedu, Gershim Asiki, Damazo T. Kadengye.

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
