## [Decision Letter · Decision Letter 0]

8 Aug 2023

PONE-D-22-20839A multi-state transition model for child stunting in two urban slum settlements of Nairobi: a longitudinal analysis, 2011-2014.PLOS ONE

Dear Dr. Iddi,

Thank you for submitting your manuscript to PLOS ONE. After careful consideration, we feel that it has merit but does not fully meet PLOS ONE’s publication criteria as it currently stands. Therefore, we invite you to submit a revised version of the manuscript that addresses the points raised during the review process.

We look forward to receiving your revised manuscript.

Kind regards,

Pratap Chandra Mohanty, Ph.D.

Academic Editor

PLOS ONE

Journal Requirements:

We sincerely acknowledge those who contributed to the establishment of the NUHDSS, especially Alex Ezeh and

Eliya Zulu. We also acknowledge funding support for the NUHDSS received from a number of donors including the

Rockefeller Foundation (USA), the Wellcome Trust (UK), the William and Flora Hewlett Foundation (USA), Comic

Relief (UK), the Swedish International Development Cooperation (SIDA) and the Bill and Melinda Gates

Foundation (USA).

5. One of the noted authors is a group or consortium Nairobi Urban Health and Demographic Surveillance System. In addition to naming the author group, please list the individual authors and affiliations within this group in the acknowledgments section of your manuscript. Please also indicate clearly a lead author for this group along with a contact email address.

Reviewers' comments:

Reviewer's Responses to Questions

**Comments to the Author**

1. Is the manuscript technically sound, and do the data support the conclusions?

Reviewer #1: Yes

Reviewer #2: Partly

2. Has the statistical analysis been performed appropriately and rigorously? 

Reviewer #1: Yes

Reviewer #2: Yes

3. Have the authors made all data underlying the findings in their manuscript fully available?

Reviewer #1: Yes

Reviewer #2: Yes

4. Is the manuscript presented in an intelligible fashion and written in standard English?

Reviewer #1: Yes

Reviewer #2: Yes

5. Review Comments to the Author

Reviewer #1: I read through the paper, and I must admit that this paper is a critical piece of missing information that the field of nutrition, public health and stunting requires: on 'mechanism of injury (pathophysiology and causation in public health" on stunting as a marker of 'disadvantage'. I read the paper with this lens. It covers all the ground well, including the methodology, which is fascinating as this type of analysis is only done in computer science fields--so this is an innovative and creative way of analysing the problem, and then quantifying it in a manner that makes public health sense.

Reviewer #2: Dear authors,

Thank you for this very interesting and important piece of work. The problem of stunting is a considerable challenge and your efforts to establish some of its determinants among vulnerable people in Kenya should be lauded. There are a number of issues with the manuscript that needs addressing - please see my specific comments below.

Introduction

1. The description of stunting as it pertains to Height-for-Age Z-scores also appears in the Methods, where I feel it is more suited. Please consider changing "It is measured by...are considered stunted." to "Stunting is defined as a child who has low height for his or her age.", and then providing the technical details pertaining to Z-scores and standard deviations to the Methods.

2. "It can further lead to irreversible brain damage." The cited paper by Cooper et al. does not say this.

3. Sentence beginning with "The past two decades have seen..." - It is not clear what is meant by "with twice as much declining rates". Please rephrase this for clarity.

4. Sentence beginning with "Since stunting mostly..." - Please change "complementary feed" to either "complementary feeding" or "complementary food".

5. Challenges around complementary feeding and breastfeeding (largely qualitative considerations) as well as MMS (largely high costs) need to be highlighted here for balance. This also supports in illustrating later findings around feeding, maternal education, and wealth.

6. Sentence beginning with "Lack of healthcare, sanitation..." - Please change "maternal education, urban / rural residency" to "maternal education, and urban or rural residency".

7. Sentence beginning with "Studies conducted in..." - Please consider changing the archaic "environs" to "environments".

8. Sentence beginning with "However, stunting prevalence is..." - Please change "low socioeconomic status (SES) households" to "low socioeconomic status (SES) urban households".

9. Please revise the way findings from references 32 and 33 are presented in the sentence "Among this population, glaring..." Ref 32 makes a very different comparison. i.e., between poor rural and rich urban populations; Ref 33 also makes comparisons around wealth exclusively, not urban/rural setting.

10. Sentence beginning with "For instance, between..." - Please change to "For instance" to "In addition". For instance does not make sense in this context as the statement does not follow on from the previous comparative works.

Materials and methods

11. Sentence beginning with "The NUHDSS is a..." - Please check the spelling of "Koroogcho".

12. Sentence beginning with "Since 2002, the..." - Please remove "nutrition" and "vaccination" from the brackets listing health outcomes. These are exposures and should be listed as such. The "other data" in the next sentence are social determinants of health and could be identified as such, if the authors so choose.

13. Please write out unitless counts below ten; e.g., "across 6 Health and Demographic Surveillance Systems", "at most 3 per child per year". "The outcome has 4 levels". There may be others not listed here.

14. Please specify the month of study commencement in 2010 and the month the study ended in 2014. Does this correspond to the 40 months in the transition probability diagram in Figure 3? If so, please make a statement that the study duration was 40 months.

15. Please indicate the ages of children included in the full sample of 3419 children.

16. Sentence beginning with "Among the various data..." - Please remove "age" from the brackets listing anthropometric measures and list this as a separate dimension.

17. The subset of 692 children is difficult to reconcile with 2889 observations in Table 1. Please indicate how these children are represented in these tables over time - presumably, children had to have follow-up visits (a maximum of three visits are listed earlier) to provide observations around state transitions; how is it possible that 692 full anthropometric datasets could be obtained if only n=117 children were observed in 2014 (Table 1)?

18. The discrepancy between 2889 observations in Table 1 and 2197 observations in Table 2 needs clarification.

19. Please justify or provide a reference to substantiate why children aged 0 to 3 years were included. For example, https://www.mattioli1885journals.com/index.php/actabiomedica/article/view/11346 suggests stunting affects development up to the age of 4 years - would it not have been more comprehensive to include a broader age range?

20. Please change "...household socio-economic data was obtained..." to "...household socio-economic data were obtained..."

21. As mentioned in the Introduction comments, the Methods section does a good job of describing the measurement of stunting using HAZ. Please simplify the Introduction section and using some detail from there in the Methods section - in particular, the description of Z-scores as standard deviations from the median growth standard is missing here.

22. There is very likely (multi)collinearity between several of the explanatory variables - while it is understood that this does not explicitly change predictive value of single explanatory variables, and acknowledged that the Akaike information criterion was used in the model, this does need to be stated. The reader needs to be aware that many of these variables are correlated; sections in the Discussion section needs to take into account that suggested policy changes for one element may have an influence on many.

23: IMPORTANT: Were ALL explanatory factors measured again at EVERY VISIT where anthropometric information was obtained? Specifically, were 'fixed' characteristics such as maternal education, exclusive breastfeeding, parental marital status measured at every visit? If not, the rationale for investigating back-transition (state improvement) as it pertains to these characteristics does not hold - for example, if maternal post-primary education (measured only once) is correlated with back-transition, there can be no conclusion that >increased< maternal education has a role to play in reducing stunting, as the level of education was constant (also while the child was transitioning to poorer states). Furthermore, this approach will likely result in a high chance of spurious associations, many of which may be driven by other collinear factors.

24. A reference to Figure 1 is needed at the start of the Statistical Analysis section.

25. The terse mathematical description refers to state space as [uppercase] S initially, then [lowercase] s throughout the rest of the section and, indeed, paper.

Results

26. "Table 1 shows the percentage of children who transitioned from one stunting state to another." Should this be Table 2?

27. "Generally, we observed that fewer...also when back transitioning." Please revise this sentence as the meaning is not clear at all.

28. Transitions that had percentages less than 10% were not considered in the model. These categories still represent upwards of 70 children. Please provide a reference or rationale for this decision. While I do understand that this may be a question of face validity, it is worrying that predictors for the transition we would MOST like to avoid (normal to severe) have not been explored, however rare it may be in the real world.

29. It is not clear why Figure 3 presents transition probabilities for the state transitions representing less than 10%. If these were not included in the model, why were TPs calculated and presented?

30. "On average, children spend about 2.4 (95% CI [1.1, 5.4]) months...and 2.27 (95% CI [1.1, 4.9]) months, respectively." Where are these values coming from? If these are additional model outputs, please provide them as accompanying or supplementary tables.

31. As mentioned previously, please indicate if the 40 months in Figure 3 corresponds to the duration of the study.

32. The entire section 3.2.2. (Factors associated with each transition state) is extremely lexically dense and difficult to digest. I would suggest further subdividing these sections with sub-headings - consider adding headings for (1) explanatory child characteristics, (2) explanatory maternal/parental characteristics, (3) explanatory social determinants of health? Alternatively, these results may also be easier to consume in a table, perhaps arranging from the factor posing the greatest to the smallest risk?

33. It is not clear why Table 4 is not referred to in the section detailing 'Normal to marginally stunted and backwards transitions"

Discussion and Conclusion

34. The first paragraph of this section is largely a repetition of the results. This paragraph would benefit from a discussion of the possible mechanisms underlying these findings.

35. "This is further confirmed by Brown, et. al." This reference could not be found; it is also noted that it is very old. Please try to find a more recent paper to substantiate this statement. If not available, please provide the full citation for Brown 1999 in the reference list.

36. The findings of the study by Rakotomanana et. al. that are presented are not the same as the present study. The present study finds rapid state transitions at 0-5 months as juxtaposed with older ages; Rakotomanana and colleagues found that stunting increased with age.

37. It is not clear what the findings of the study by Emily et. al. are - children in their second year of life have a higher likelihood to be underweight and stunted as compared to what?

38. Please change "food insecured" and food secured" to "food insecure" and "food secure", respectively.

39. Sentence beginning with "The effect of parent marital status..." - Please change "largely significant" to something like "notable". This finding is not significant if it is not statistically significant.

40. IMPORTANT: From "Regarding the effect of mother's educational status...odds of child stunting in Malawi, Tanzania, and Zimbabwe." Back-transitioning is NOT the same as never having been stunted. The findings from other studies suggesting that maternal education results in lower odds of being stunted indicates a protective association of maternal education. As stated before, back-transition associated with maternal education can ONLY show the latter as a protective factor if maternal education was measured every time anthropometric measures were taken, and maternal education was increasing as stunting was decreasing.

41. I find "menace" to be a strange, emotive and redundant addition to "stunting". I'll defer to the editor as this is a style choice, but would prefer if it was not framed in this way.

42. IMPORTANT: It is fairly disorienting to paper where no reflections regarding the strengths and limitations of the study are presented as part of the Discussion. Please consider adding this to the manuscript.

6. PLOS authors have the option to publish the peer review history of their article (what does this mean?). If published, this will include your full peer review and any attached files.

Reviewer #1: **Yes: **Shuaib Kauchali

Reviewer #2: **Yes: **Amanda Salomé Brand

---

## [Author Response · Author response to Decision Letter 0]

14 Sep 2023

We have carefully reviewed the reviewers comments and responded to each of them. The revised manuscript has improved substantially after considering all the comments from the reviewers. A point-by-point response to reviewers comments have been upload to the system.

---

## [Decision Letter · Decision Letter 1]

5 Dec 2023

PONE-D-22-20839R1A multi-state transition model for child stunting in two urban slum settlements of Nairobi: a longitudinal analysis, 2011-2014.PLOS ONE

Dear Dr. Iddi,

Thank you for submitting your manuscript to PLOS ONE. After careful consideration, we feel that it has merit but does not fully meet PLOS ONE’s publication criteria as it currently stands. Therefore, we invite you to submit a revised version of the manuscript that addresses the points raised during the review process.

We look forward to receiving your revised manuscript.

Kind regards,

Engelbert A. Nonterah, MD, PhD

Academic Editor

PLOS ONE

Journal Requirements:

Reviewers' comments:

Reviewer's Responses to Questions

**Comments to the Author**

1. If the authors have adequately addressed your comments raised in a previous round of review and you feel that this manuscript is now acceptable for publication, you may indicate that here to bypass the “Comments to the Author” section, enter your conflict of interest statement in the “Confidential to Editor” section, and submit your "Accept" recommendation.

Reviewer #2: (No Response)

Reviewer #3: (No Response)

2. Is the manuscript technically sound, and do the data support the conclusions?

Reviewer #2: Yes

Reviewer #3: Yes

3. Has the statistical analysis been performed appropriately and rigorously? 

Reviewer #2: Yes

Reviewer #3: Yes

4. Have the authors made all data underlying the findings in their manuscript fully available?

Reviewer #2: Yes

Reviewer #3: Yes

5. Is the manuscript presented in an intelligible fashion and written in standard English?

Reviewer #2: Yes

Reviewer #3: Yes

6. Review Comments to the Author

Reviewer #2: Dear authors,

Thank you very much for engaging so constructively with the previous round of feedback, this is much appreciated. The paper is looking great and is, in my view, more clear and accessible for the reader following your revisions. Please see a few additional comments below:

Comment Introduction #1 (previously comment #1): Thank you very much for the revision of the description around stunting in the Introduction and Materials and methods sections. Two small follow-up comments in the Materials and Methods section - there is a typo in Line 124 (please change ‘HAV’ to ‘HAZ’); also, please consider adding a sentence explaining why there is a ‘marginal’ stunting category even though it does not meet the strict WHO definition of stunting (< 2 SDs below the median). My suggestion for the latter would be to add the following: ‘…and severely stunted (HAZ < -3). While marginal stunting does not fall into the strict definition of stunting, it cannot be ignored as it represents a barrier to thriving (Tredoux, C., Dawes A., Mattes, M. (2022) Thrive by Five Index 2021 Technical Report (Revised July 2022). University of Cape Town and Innovation Edge, Cape Town.) In addition to the stunting state…’

Comment Materials and methods #2 (previously comment #19): Thank you very much for the clarification that children between the 0 to 3 years were recruited to allow sufficient time to observe their outcomes before their 5th year and that, therefore, outcomes of children between 4 and 5 were observed for those with late entry year to the study. Would it be possible to add a sentence to support the reader with this understanding?

Comment Materials and methods #3 (previously comment #22): The authors are thanked for considering the comment regarding collinearity and for further statistical calculations to estimate this. Without the numeric values, however, it is not clear now weak these associations are – please note some sources consider a Cramér's V of >0.10 moderate and >0.15 strong (Dai, J., Teng, L., Zhao, L., & Zou, H. (2021). The combined analgesic effect of pregabalin and morphine in the treatment of pancreatic cancer pain, a retrospective study. Cancer Medicine, 10(5), 1738–1744. https://doi.org/10.1002/cam4.3779). It is acknowledged that other sources do define V < 0.2 as weak; in the presence of this conflicting advice, it is recommended that exact values are presented and the reference defining these as ‘weak’ cited.

Comment Results #4 (previously comment #27): The authors are thanked for revising this section of text for clarity. It is not apparent that the revision is reflecting the previously intended meaning (interpreted as stating that very few children skipped a consecutive state of severity as they transitioned, either forward or backward, through states). It appears from the reduction of Figure 1 to Figure 2 in the revised manuscript as though these events were less than 10% in each case and therefore not considered in the model. If this is not the case, it is up to the authors whether they want to retain the more general current statement, or reconsider.

Comment Results #5, Line 282-284: ‘The risk of back transitioning from the moderately stunted to marginally stunted state for children in moderately food secured and severely food secured households are, respectively, 2.9 and 1.7 times the risk for children in food secured households’ does not say the same as the Discussion: ‘…we observed that children living in households that experienced moderate to severe food insecurity were at an increased risk to transition into a moderately stunted state from a marginally stunted state, relative to those living in food secure households.’ Please revise lines 282-284 for clarity – it is not clear what ‘moderately food secured’ and ‘severely food secured’ means in relation to ‘food secured’.

Comment Results #6, Line 293-295: ‘…when in a marginally stunted state, children whose mothers have attained primary or post primary education are, respectively, 2.85 times and 3.6 more likely to transition into a moderately stunted state compared to children with a less than primary educated mothers.’ does not say the same as the Discussion: ‘…the results indicated that children in a moderately stunted state were more likely to back-transition to marginally stunted state if they have mothers with primary and post-primary education. This finding is consistent with a study by Abuya et. al. [35], who observed that children born to primary educated mothers were at a significantly lower odds of being stunted relative to mothers with no primary education.’ Please revise lines 293-295, which currently implies that children of mothers with primary+ education are at higher risk of transitioning from marginally to moderately stunted.

Reviewer #3: Thank you for submitting this great piece of work for publication in this journal. You have have chosen a topic of immense public health importance, not only to Kenyan society, but to the entire tropical environment. The communication level and use of English language is excellent. I believe that this manuscript is suitable for publication in this journal but I also have a few comments and observations which require further attention from the authors.

I am not sure if the title of the article (A multi-state transition model for child stunting in two urban slum settlements of

Nairobi: a longitudinal analysis, 2011-2014) correctly describes the contents. As it is, it suggests that the main outputs of this work are some models developed to predict or quantify stunting while in reality, the authors used modeling techniques to measure and classify stunting into 4 degrees of severity. I believe that the authors can improve the understanding of the title by making some adjustment to it.

I am also a bit concerned about the selection of the subset of 692 children as study participants because they were the ones with complete anthropometric data, and whose household socioeconomic data were obtained from the NUHDSS. This set of children may not be representative of all children from these slums. The reasons why they have complete anthropometric data and household socioeconomic data may also make them different from the other children without complete information. Could the authors also performed statistical analysis of key potentially confounding factors between these children with complete information (research participants) and those with incomplete information (excluded from participation) to see how similar or different they are? Again related to study design, how were the 'factors associated with transition between stunting states' determined or chosen?

In your discussion, I expected to see how the authors discuss the positive or negative effects (if any) between one stunting state to another in order to highlight the significance of these transitions. Is it feasible to add something on that?

Finally, I also expect the authors to emphasize on the need for early screening of stunting before the age of 6 months since children between the age of 0 - 5 months are more likely to transitioned from poorer stunting states to better statuses.

7. PLOS authors have the option to publish the peer review history of their article (what does this mean?). If published, this will include your full peer review and any attached files.

Reviewer #2: **Yes: **Amanda Salomé Brand

Reviewer #3: **Yes: **Salisu M. Ishaku

---

## [Author Response · Author response to Decision Letter 1]

20 Dec 2023

We thank the reviewers for the comments and suggestion. A point-by-point response to all comments have been upload to the system.

---

## [Editor Report · Decision Letter 2]

6 Feb 2024

Utilizing a multi-stage transition model for analysing child stunting in two urban slum settlements of Nairobi: a longitudinal analysis, 2011-2014

PONE-D-22-20839R2

Dear Dr. Samuel Iddi,

We’re pleased to inform you that your manuscript has been judged scientifically suitable for publication and will be formally accepted for publication once it meets all outstanding technical requirements.

Kind regards,

Engelbert A. Nonterah, MD, PhD

Academic Editor

PLOS ONE
---

## [Editor Report · Acceptance letter]

16 Feb 2024

PONE-D-22-20839R2 

PLOS ONE

Dear Dr. Iddi, 

I'm pleased to inform you that your manuscript has been deemed suitable for publication in PLOS ONE. Congratulations! Your manuscript is now being handed over to our production team.

Kind regards, 

on behalf of

Dr. Engelbert Adamwaba Nonterah 

Academic Editor

PLOS ONE